# *Lactobacillus helveticus* HY7804 Modulates the Gut–Liver Axis to Improve Metabolic Dysfunction-Associated Steatotic Liver Disease in a Mouse Model

**DOI:** 10.3390/ijms26083557

**Published:** 2025-04-10

**Authors:** Hyeonji Kim, Hye-Jin Jeon, Ji-Woong Jeong, Kippeum Lee, Hyeonjun Gwon, Daehyeop Lee, Joo-Yun Kim, Jae-Jung Shim, Jae-Hwan Lee

**Affiliations:** R&BD Center, hy Co., Ltd., 22, Giheungdanji-ro 24beon-gil, Giheung-gu, Yongin-si 17086, Republic of Korea; skyatk94@gmail.com (H.K.); 10003012@hy.co.kr (H.-J.J.); woongshow@hy.co.kr (J.-W.J.); 10002903@hy.co.kr (K.L.); hjgwon@hy.co.kr (H.G.); flywhy7@hy.co.kr (D.L.); jjshim@hy.co.kr (J.-J.S.); jaehwan@hy.co.kr (J.-H.L.)

**Keywords:** *Lactobacillus helveticus*, MASLD, intestinal inflammation, tight junction, gut microbiota

## Abstract

Metabolic dysfunction-associated steatotic liver disease (MASLD) is the most common type of liver disease worldwide. In a previous study, we confirmed that *Lactobacillus helveticus* HY7804 (HY7804) improves MASLD by suppressing the expression of mRNAs encoding genes related to hepatic lipogenesis, inflammation, and fibrosis in model mice. Here, we evaluated the ability of HY7804 to restore intestinal barrier function and modulate the gut microbiota, as well as improve MASLD symptoms. Mice fed an MASLD-inducing diet for 7 weeks received HY7804 (10^9^ CFU/kg/day), the Type strain, or positive control (Pioglitazone) during the same period. HY7804 alleviated physiological (*p* < 0.001) and blood biochemical indicators and reduced MASLD activity scores (*p* < 0.05) on histological analysis. In addition, HY7804 increased the expression of genes related to fatty acid oxidation (*p* < 0.001); decreased the expression of apoptosis-related genes (*p* < 0.001); rescued the expression of tight junction (TJ)-related genes (*p* < 0.05); and suppressed the expression of pro-inflammatory cytokines and TLR4/MyD88/NF-κB signaling (*p* < 0.01) in the intestine. Finally, HY7804 modulated the composition of the gut microbiota in MASLD-induced mice. HY7804 increased the abundance of MASLD-suppressive *Bacteroidaceae* and *Bacteroides*, which positively correlated with the expression of TJ- and fatty acid oxidation-related genes. By contrast, HY7804 decreased the abundance of bacteria related to the progression of MASLD, including *Cloastridaceae*, *Clostridium*, *Streptococcaceae*, *Lactococcus*, and *Lachnospiraceae*, which correlated with intestinal immune responses and MASLD symptoms. In conclusion, *L. helveticus* HY7804 may be suitable as a functional supplement that alleviates MASLD symptoms and improves intestinal health.

## 1. Introduction

Metabolic dysfunction-associated steatotic liver disease (MASLD, previously referred to as non-alcoholic fatty liver disease (NAFLD)), a hepatic manifestation of metabolic syndrome and a major cause of chronic liver disease, has an estimated global prevalence of nearly 30% [1]. The disease takes various forms, from simple steatosis to non-alcoholic steatohepatitis, liver fibrosis, cirrhosis, and hepatocellular carcinoma [2,3]. Generally, MASLD is closely associated with obesity-related metabolic disorders such as insulin resistance and metabolic syndrome, although there is also a link to cardiovascular disease, cardiac disease, chronic kidney disease, and intestinal bowel disease [4,5,6,7].

The “gut–liver axis”, which refers to bidirectional interactions between the liver and the gut/microbiota, plays a critical role in the development and progression of MASLD [8]. The pathogenesis of MASLD has a correlation to disruption of the intestinal barrier, tight junction alterations, and disturbance of the gut microbiota [9]. Intestinal epithelial cells maintain the integrity of the intestinal barrier [10]. TJ proteins form complexes at the junctions between epithelial cells; disruption of TJ complexes leads to increased permeability of the intestinal barrier [9]. However, the disruption of TJ complexes leads to increased intestinal barrier permeability [11,12]. Interference with dietary fat metabolism also causes changes in intestinal permeability by disrupting the function of the intestinal barrier [13]. Previous studies show that in MASLD patients, the expression of zonular occludin-1 and occludin, which are representative TJ proteins, is decreased, and intestinal permeability is greater than that in healthy individuals [14]. Furthermore, increased intestinal permeability due to MASLD is closely related to intestinal inflammation [15]. A recent study revealed that the gut microbiota plays a role in intestinal permeability and host immune deficiency [9,15]. In addition, disruption of the gut microbiota, as well as bacterial metabolic products, are detrimental to the liver, contributing to the generation of hepatic steatosis and fibrosis [16].

Probiotics, defined by the Food and Agriculture Organization (FAO)/ World Health Organization (WHO) as “live microorganisms, which when administrated in adequate amounts confer a health benefit on the host” [17], have various beneficial effects, including maintenance of the gastrointestinal tract and the intestinal microbial balance; moreover, they also have beneficial effects in the context of liver diseases, and skin and mental functions [18,19,20,21]. Previously, we showed that *Lactobacillus helveticus* HY7804 (HY7804) alleviates MASLD in a mouse model by inhibiting genes related to lipogenesis, inflammation, and fibrosis in liver tissues [22]. In addition, the administration of HY7804 activated the expression of β-oxidation-related genes such as acyl-CoA oxidase 1 (*Acox1*), carnitine palmitoyltransferase 1A (*Cpt1a*), and peroxisome proliferator-activated receptor alpha (*Pparα*). Additionally, HY7804 improves histological symptoms such as hepatic steatosis, lobular inflammation, and hepatocyte ballooning. However, despite evidence that HY7804 improves MASLD symptoms, the effects of HY7804 on the “gut–liver axis”, including colonic inflammation, intestinal barrier function, and composition of the gut microbiota have not been studied.

The purpose of this study was to use an MASLD mouse model to investigate how *Lactobacillus helveticus* HY7804 improves MASLD symptoms via the gut–liver axis, restores TJ functions, and suppresses intestinal inflammation. In addition, we investigated how HY7804 affects the diversity of the intestinal microbial community. Finally, we examined the correlation between histological scores, expression of hepatic/colonic genes, and the composition of gut microbiota to determine the effects of changes in the intestinal flora on MASLD symptoms.

## 2. Results

### 2.1. Effects of HY7804 on Physiological Indicators in MASLD-Induced Mice

To study whether HY7804 intake improves MASLD, we investigated the effects of HY7804 on mice fed an MASLD-inducing diet. Pioglitazone (PIO), used as a positive control in this animal study, is a potential treatment for MASLD [23]. We measured body weight changes in mice and dietary intake during the experiments (Figure 1A and Appendix A). Figure 1A shows that there were significant differences in body weight between the mouse groups during the experimental period. The body weight of the MASLD-induced group (*p* < 0.001) was greater than that of the control group. Body weight in the PIO, HY7804, and Type strain (*Lactobacillus helveticus* type strain KCTC 3545) intake groups was significantly lower than that of the MASLD-induced group (*p* < 0.001). Figure 1B shows the food efficiency ratio (FER, %) of each group, calculated as body weight gain (g/day)/dietary intake (g)} × 100. The FER of the MASLD group was significantly higher (68.12 ± 8.54 g, *p* < 0.001) than that of the control group (36.36 ± 5.22 g). Treatment with HY7804 led to a significant reduction in the FER (to 49.12 ± 12.44 g; *p* < 0.01) compared with that of the MASLD group. Both PIO and the *L. helveticus* type strain also led to a reduction in the FER, but the difference from that of the MASLD group was not significant.

Figure 1C–E shows the liver, epididymal fat, and inguinal fat masses of the mice in all the groups. The mass of these tissues in the MASLD-induced group was higher than that in control mice (*p* < 0.001). The mass of these tissues from mice in the PIO, HY7804, and Type strain groups was significantly lower than in the MASLD group (*p* < 0.001). Figure 1F–H shows the ratio of liver/epididymal fat/inguinal fat tissue weight to body weight, respectively. The trend is the same as that for tissue weight. The ratio of tissue/body weight in the MASLD group was significantly higher than that in the control group (*p* < 0.001). The administration of PIO, HY7804, or the Type strain led to a significant reduction in the liver/epididymal fat/inguinal fat tissue to body weight ratios, respectively, compared with that of the MASLD-induced group (PIO and HY7804 groups, *p* < 0.001; Type strain group, *p* < 0.01). Thus, HY7804 was efficient at inhibiting increases in the weight of liver/epididymal fat/inguinal fat tissue (HY7804 > PIO > Type strain).

### 2.2. Effects of HY7804 on Blood Biochemical Analysis

Next, we analyzed the levels of alanine aminotransferase (ALT) and aspartate aminotransferase (AST) in serum samples from all the groups (Figure 2A,B). The levels of ALT and AST were significantly higher in the MASLD-induced mice than in the control mice (*p* < 0.001). The administration of PIO, HY7804, or the Type strain led to a significant reduction in the ALT and AST levels relative to those in the MASLD group (ALT, *p* < 0.05; AST, *p* < 0.01). As shown in Figure 2C–F, the elevated triglyceride, total cholesterol, fasting glucose, and LDL-cholesterol levels induced by MASLD (*p* < 0.001) fell significantly after treatment with PIO, HY7804, or the Type strain (*p* < 0.001). Figure 2G shows that the level of HDL-cholesterol in the MASLD group was significantly higher than that in the control group (*p* < 0.01). The HDL-C levels in the PIO and HY7804 groups were slightly higher than those in the MASLD group, but the difference was not significant. Treatment with the Type strain resulted in HDL-C levels similar to those in the MASLD group.

### 2.3. Effects of HY7804 on Hepatic Histological Analysis

Next, we investigated whether HY7804 affects liver tissue morphology and histology. Figure 3A shows the morphology of the liver tissue harvested from each group. The induction of MASLD caused the liver to increase in size and become whitened. Visual inspection indicated that PIO, HY7804, and the Type strain rescued this morphology. Of the treated groups, the HY7804 group’s liver morphology most closely resembled that of the control group. Figure 3B shows that an MASLD-induced diet causes the accumulation of hepatic fat, which is reversed by the administration of PIO or probiotics.

Next, we assessed the NAS (MASLD Activity score), which includes steatosis, lobular inflammation, and hepatocyte ballooning, as observed on hematoxylin and eosin (H&E)-stained slides. Figure 3C shows that the high steatosis grade in mice fed an MASLD-inducing diet decreased in all the treated groups; however, among these, only the HY7804 group had a significantly lower steatosis grade than the MASLD group (*p* < 0.05). The high grade of lobular inflammation and hepatocyte ballooning in the MASLD-induced mice was also lower in all the treatment groups (Appendix A). Additionally, the NAS (calculated by summing grade of steatosis, lobular inflammation, and hepatocyte ballooning) was significantly higher in the MASLD-induced mice (*p* < 0.001) compared with the control group, but fell significantly upon the administration of HY7804 (*p* < 0.05).

### 2.4. Effects of HY7804 on Hepatic Gene Expression

To determine the effects of HY7804 on the expression of hepatic genes, we examined the levels of mRNA encoding genes related to lipogenesis, lipid oxidation, and apoptosis. Figure 4A–C shows that the expression of mRNAs encoding *Fasn* (7.33-fold), *Srebp-1c* (8.31-fold), and *C/ebpα* (1.27-fold) increased in mice fed an MASLD-inducing diet (*p* < 0.001). Treatment with PIO, HY7804, or the Type strain suppressed the expression of these genes significantly. The expression of *Pparγ* in the MASLD group tended to be higher (3.40-fold, *p* < 0.001) than in the control group (Figure 4D). HY7804 led to significantly lower expression of *Pparγ* than that observed in the MASLD group (1.58-fold, *p* < 0.05). PIO and the Type strain decreased *Pparγ* expression slightly, but the difference between them and the MASLD group was not significant. In other words, HY7804 showed an inhibitory effect on hepatic lipogenesis at a higher level than other groups.

Next, we examined the expression of the lipid oxidation-related genes *Cpt1a* and *Ppagc1a* (*PGC1a*). The expression of *Cpt1a* and *PGC1a* mRNA showed similar patterns (Figure 4E,F); expression of both genes fell significantly in the MASLD group (*Cpt1a*, 0.40-fold, *p* < 0.001; *PGC1a*, 0.41-fold, *p* < 0.05), but was restored in the treatment groups. In particular, treatment with HY7804 increased mRNA expression to the greatest extent when compared with that in the MASLD group (*Cpt1a*, 1.06-fold; *PGC1a*, 1.42-fold, *p* < 0.001). By contrast, expression in the Type strain group was slightly higher than that in the MASLD group, but the difference was not significant.

In addition, we investigated whether HY7804 affects hepatic apoptosis-related markers such as the *Bax*/*Bcl-2* ratio in the liver tissues (Figure 4G). The *Bax*/*Bcl-2* ratio in the MASLD-induced diet group was higher than that in the control group (by 1.39-fold, n.s). The administration of PIO, HY7804, and the Type strain reduced the *Bax*/*Bcl-2* ratio significantly to 1.00-fold (*p* < 0.05), 0.82-fold (*p* < 0.001) and 0.85-fold (*p* < 0.001), respectively, compared with the MASLD group. Thus, HY7804 reduces the expression of lipogenesis-related genes, upregulates the expression of β-oxidation-related genes, and modulates apoptosis-related markers more effectively than the Type strain.

### 2.5. Effects of HY7804 on Expression of TJs and Inflammation-Related Genes in Colon Tissue

Next, we investigated whether HY7804 affects the expression of TJ-related genes in colon tissue. As shown in Figure 5A, the MASLD-induced downregulation of *Ocln* expression (0.70-fold, *p* < 0.01) was reversed significantly by PIO (1.00-fold, *p* < 0.01) and HY7804 (0.97-fold, *p* < 0.05), both of which returned levels of *Ocln* almost to those observed in the control group. The level of *Ocln* in the group treated with the Type strain also increased, but not significantly. Figure 5B shows that the expression of *Cldn1* in the MASLD group (0.30-fold, *p* < 0.001) was significantly lower than that in the control group, whereas that in the PIO, HY7804, and Type strain groups was 0.68-fold (*p* < 0.01), 0.53-fold (*p* < 0.05), and 0.47-fold (n.s) higher, respectively, than that in the MASLD group. These data suggest that HY7804 restores the expression of TJ-related genes more effectively than the Type strain.

Next, we assessed the expression of genes encoding pro-inflammatory cytokines and components of the TLR/MyD88/NF-κB pathway. In the group fed an MASLD-induced diet, the relative expression of *Il6*, *Il1β*, and *Tnfα* was significantly higher (by 1.99-fold (*p* < 0.01), 3.04-fold (*p* < 0.001), and 1.42-fold (*p* < 0.05), respectively) than that in the control group (Figure 5C–E). By contrast, the relative expression of *Il6*, *Il1β*, and *Tnfα* in the HY7804 group was 1.02-fold (*p* < 0.01), 0.94-fold (*p* < 0.001), and 0.92-fold (*p* < 0.01), respectively. HY7804 reduced the expression of *Il6*, *Il1β*, and *Tnfα* to levels similar to those observed in the control group. The Type strain also reduced the expression of these cytokine genes compared with the MASLD group, but only the difference in *Il1β* expression was significant.

As illustrated in Figure 5F–H, the relative expression of mRNAs encoding *Tlr4*, *Myd88*, and *Nfκb1* was upregulated by the MASLD-induced diet (by 1.74-fold (n.s), 1.22-fold (*p* < 0.01), and 1.13-fold (n.s), respectively) compared with the control group. The administration of HY7804 suppressed expression significantly (*Tlr4*, 0.45-fold; *Myd88*, 0.96-fold; *Nfκb11*, 0.78-fold, *p* < 0.01). The increase in expression of *Tlr4*, *Myd88*, and *Nfκb1* mRNA after MASLD induction fell slightly after the oral administration of PIO and the Type strain, but the differences were not significant. Taken together, these results show that HY7804 exerts a greater suppressive effect on the expression of genes encoding pro-inflammatory cytokines and components of the TLR/MyD88/NF-κB signaling pathway than other groups.

### 2.6. Effects of HY7804 on the Composition of the Gut Microbiota in the MASLD-Induced Mouse Model

To analyze changes in the gut microbiota community induced by MASLD and the effects of HY7804, we sequenced the V3-V4 region of the 16S rRNA gene using the Illumina MiSeq platform (Appendix A). First, we analyzed the α-diversity of the gut microbiota (i.e., the Shannon and Simpson indices; Figure 6A). The Shannon and Simpson indices decreased in mice fed an MASLD-inducing diet, but increased after the intake of HY7804 or the Type strain (= Type_helveticus, *Lactobacillus helveticus* type strain KCTC 3545) (Shannon, *p* = 0.02735; Simpson, *p* = 0.016681). By contrast, PIO did not alter the reduced α-diversity indices in MASLD mice. Figure 6B shows the β-diversity, according to Bray–Curtis dissimilarity, between each group. A principal coordinate analysis (PCoA) revealed that the intestinal bacterial composition of the MASLD mice was different from that of the control group. Also, composition in the HY7804 and Type strain groups was distinct from that in the MASLD group. Next, we investigated the abundance of phylum within the gut microbiota (Figure 6C). The relative abundance of *Bacteroidota* (86.16% vs. 35.69%) and *Pseudomonadota* (1.75% vs. 0.09%) in the mice fed an MASLD-induced diet was higher, and that of *Bacillota* (0.34% vs. 8.44%) lower, than that in the control group. The administration of HY7804 reduced the abundance of *Bacteroidota* and *Pseudomonadota* to 67.86% and 0.50%, respectively, and increased the abundance of *Bacillota* to 15.00%. Figure 6D,E show an analysis of the intestinal microbiota at the family and genus levels. At the family level, the relative abundance of *Bifidobacteriaceae*, *Rinenellaceae*, *Tannerellaceae*, *Bacteroidaceae*, and *Erysipelotrichaceae* was lower in the MASLD-induced group than in the control group. By contrast, the abundance of *Clostridaceae*, *Steptococcaceae*, *Lachnospiraceae*, and *Lactobacillaceae* in the MASLD group was higher than that in the control group; however, these trends tended to be reversed by the administration of PIO and probiotics (i.e., HY7804 and Type_helveticus). At the genus level, an MASLD-induced diet reduced the abundance of *Bifidobacterium*, *Alistipes*, *Parabacteroides*, *Bacteroides*, and *Faecalibaculum* compared with the control group, whereas the relative abundance of *Clostridium*, *Lactococcus*, *Dorea*, and *Lactobacillus* was higher than in the control group. At the family level, PIO and probiotics reversed these changes in the composition of the intestinal microbiota. In particular, the administration of HY7804 modulates the relative abundance of *f_Bacteoidaceae*, *g_Bacteroides*, *f_Cloastridaceae*, *g_Clostridium*, *f_Streptococcaceae*, *g_Lactobacoccus*, and *f_Lachnospiraceae* to levels similar to those in the control group.

### 2.7. Correlation Between Relative Abundance and MASLD-Related Indicators

Finally, we examined correlations between the intestinal microbial abundance, histological indicators, and expression of hepatic/colonic genes. As illustrated in Figure 7, the expression of genes related to TJs (*Ocln* and *Cldn1*) and fatty acid β-oxidation (*Cpt1a* and *PGC1*) correlated negatively with the increased abundance of the progression of MASLD-related bacteria (*f_Clostridaceae*, *f_Steptococcaceae*, *f_Lachnospiraceae*, *f_Lactobacillaceae*, *g_Clostridium*, g_*Lactococcus*, g_*Dorea*, and *g_Lactobacillus*).

By contrast, the expression of these genes correlated positively with the abundance of f_Bifidobacteriaceae, f_Rinenellaceae, f_Tannerellaceae, f_Bacteroidaceae, f_Erysipelotrichaceae, g_Bifidobacterium, g_Alistipes, g_Parabacteroides, g_Bacteroides, and g_Faecalibaculum. In addition, these bacteria showed a negative correlation with the genes related to hepatic lipogenesis and apoptosis, histological indicators, and intestinal immune responses.

## 3. Discussion

MASLD is the most common type of chronic liver disease, with severity ranging from simple steatosis to steatohepatitis, advanced fibrosis, and cirrhosis [24]. In a previous study, we demonstrated that the *Lactobacillus helveticus* HY7804 (HY7804) alleviates MASLD symptoms by suppressing mRNA encoding genes related to hepatic lipogenesis, inflammation, and fibrosis in mice fed an MASLD-inducing diet [22]. In the present study, we show that HY7804 restores the expression of genes encoding TJ proteins, and reduces the expression of genes encoding pro-inflammatory cytokines in colon tissues. In addition, we also examined the diversity and changes in the composition of the gut microbial community to identify correlations between the gut microbiota and indicators in the MASLD-induced mouse model.

The mice were fed a diet comprising 40 kcal% fat (mostly palm oil), 20 kcal% fructose, and 2% cholesterol for 7 weeks to induce MASLD (Cat no. D09100310). PIO was used as a positive control. PIO is a potential therapy for MASLD that acts by improving fasting glucose and triglyceride levels, and by reducing insulin resistance [25]. The high fat, high fructose, and high cholesterol diet caused significant increases in body weight, FER, and mass of liver/epididymal fat/inguinal fat tissues; all of these were reduced by HY7804.

Elevated levels of ALT and AST are associated with non-specific damage to hepatocytes [26]. In general, hepatic steatosis due to MASLD has a biochemical characteristic of increased transaminase levels [27]. Indeed, we found that the serum concentrations of ALT and AST in the MASLD-induced mice were high, but fell significantly upon treatment with HY7804 (to levels similar to those after PIO treatment). MASLD is thought to result from liver metabolic disorders, which themselves result from the excessive accumulation of triglycerides and cholesterol [28,29]. In addition, increased fasting blood glucose levels may be a biomarker of severe hepatic steatosis, and are related to the degree of liver “fattiness” in patients with MASLD [30]. We confirmed that HY7804 reversed the diet-induced increases in triglyceride, total cholesterol, glucose, and LDL-cholesterol levels. Indeed, HY7804 significantly decreased these parameters to levels similar to those in the PIO group.

A histopathology analysis confirmed that HY7804 improves liver tissue morphology and MASLD biomarkers. The NAS score and steatosis significantly decreased only after the administration of HY7804. Taken together, these results show that HY7804 ameliorates MASLD from a histological perspective.

Elevated expression of hepatic lipogenesis-related genes in mice fed an MASLD-induced diet confirmed that liver steatosis occurs actively in this model. We found that the expression of mRNA encoding *Fasn*, *Srebf1*, *C/ebpα*, and *Pparγ* fell significantly after the administration of HY7804. These genes play a role in regulating de novo lipogenesis and fat accumulation in the liver. The upregulated expression of lipogenesis-related genes is associated with the progression of MASLD [31,32]. We also confirmed that HY7804 alleviated the expression of β-oxidation-related genes (*Cpt1a* and *Ppargc1a*), which decreased in mice fed an MASLD-inducing diet. Fatty acid β-oxidation is an important source of metabolic energy during high energy demand states. This metabolic state induces the release of fatty acids via the secretion of circulating mediators, thereby increasing the rate of lipolysis to generate energy [33]. Previous studies show that promoting fatty acid oxidation through the activation of *Cpt1a* and *Ppargc1a* improves MASLD [34,35]. An increase in the *Bax*/*Bcl-2* expression ratio may be a critical indicator of apoptosis in individuals with MASLD [36]. The expression of apoptosis-related genes is thought to indicate hepatocyte ballooning, a form of hepatocellular death observed in histopathology [37]. Our results imply that HY7804 improves MASLD by suppressing hepatic fat steatosis and apoptosis, and by activating fatty acid β-oxidation.

The progression of MASLD correlates with increased colonic epithelial permeability and TJ destruction [9]. In general, the function of the intestinal barrier is dependent on TJ proteins such as occludin and claudin [38]. Here, we confirmed that HY7804 restores the expression of the mRNA encoding TJ-related genes. The destruction of the intestinal barrier increases intestinal permeability and intestinal inflammatory responses [39,40]. Upregulated inflammation triggered by an MASLD-induced diet was confirmed by investigating the expression of pro-inflammatory-related genes in the colon. We found that the expression of *Il6*, *Il1β*, and *Tnfα* fell significantly after the administration of HY7804. In addition, our data show that HY7804 suppressed expression of mRNA encoding *Tlr4*, *Myd88*, and *Nfκb1*, which was raised by the MASLD-induced diet. The TLR4/Myd88/NF-κB signaling pathway is closely associated with the regulation of intestinal barrier integrity and mucosal immune responses [41,42]. Furthermore, increased TJ permeability is related to the downregulation of the TLR4/Myd88/NF-κB signaling pathway [43,44]. Thus, these results suggest that HY7804 has the potential to increase intestinal barrier functions that are disrupted by MASLD.

Our data show HY7804 significantly modulated certain markers that PIO and the Type strain did not. HY7804’s effects were largely comparable to PIO, and in certain measures, HY7804 was as effective as or even more effective than PIO. For example, HY7804 was the only treatment to significantly reduce hepatic steatosis grade and NAS (whereas PIO showed a reduction that did not reach significance in our model), indicating a potentially superior effect of HY7804 on liver fat accumulation. We also confirmed that HY7804 restored tight junction gene expression and reduced inflammatory markers in the colon to near-normal levels, similar to PIO. Especially, as previously mentioned, HY7804 significantly reduced colonic inflammatory gene expression compared to the PIO group. Furthermore, our results indicate that HY7804 outperformed the Type strain in most parameters. For instance, the HY7804 treatment led to a reduction in steatosis score and Pparγ expression, whereas the Type strain group showed improvements in these measures that did not reach significance. And, the expression level of genes encoding beta-oxidation showed a significant difference between HY7804 and Type strain. Also, HY7804 reduced colonic inflammatory genes and TLR/MyD88/NFκB pathway-related gene expression compared to the PIO group. In summary, the administration of HY7804 may be as effective in improving MASLD symptoms compared with PIO and Type strain groups.

Finally, we demonstrated that increased intestinal permeability and inflammation in the MASLD-induced mouse model was accompanied by dysbiosis of the intestinal microbial flora. We also analyzed the correlation between the gut microbiota and other biochemical indicators. Previous studies demonstrate that the diversity of the gut microbiota in MASLD-induced mice differs significantly from that in healthy mice [45,46]. In our study, HY7804 restored the α-diversity reduced by MASLD. Although the 3D PCoA plot of β-diversity demonstrated distance between the control and MASLD groups, it also showed that the HY7804 group was much closer to the CON group than the MASLD group. In addition, we show that HY7804 reduces the abundance of *Clostridium* (belonging to the *Clostridaceae* family), *Lactococcus* (belonging to the *Steptococcaceae* family), *Dorea* (belonging to the *Lachnospiraceae* family), and *Lactobacillus* (belonging to the *Lactobacillaceae* family). According to previous papers, these genera and families are more abundant in patients with MASLD than in healthy individuals [45,46,47].

Interestingly, *Lactbacillaceae* and *Lactobacillus* are representative probiotics, which improve and reduce the disruption of intestinal barrier function [48,49]. Also, the intake of *Lactobacillus* improves liver diseases [50]; however, we found that the MASLD group had a higher abundance of *Lactobacillaecae* and *Lactobacillus* than the control group. Previous studies also report a high abundance of *Lactobacillaecae* and *Lactobacillus* in MASLD patients [51,52]. The intake of high fat and high fructose diet, as well as excessive carbohydrates and calories, induces the accumulation of liver fat, resulting in symptoms of MASLD [22]. The worsening of MASLD symptoms induces the dysbiosis of the gut microbiota, which destroys gut homeostasis and, increases excessively *Lactobacillus*, thereby reducing the diversity of the intestinal flora. Here, we confirmed that HY7804 restores gut microbiota homeostasis, and returns *Lactobacillus* to control group levels, as well as increasing diversity and improving MASLD symptoms.

HY7804 reversed the effects of MASLD by increasing the relative abundance of *Bifidobacterium* (belonging to the *Bifidobacteriaceae*), *Alistipes* (belonging to the *Rinenellaceae*), *Parabacteroides* (belonging to the *Tannerellaceae*), *Bacteroides* (belonging to the *Bacteroidaceae*), and *Faecalibaculum* (belonging to the *Erysipelotrichaceae*). These genera and families are less abundant in patients with MASLD than in healthy individuals. These results suggest that the abundance of these bacteria correlates negatively with MASLD symptoms. *Alistipes* may protect against liver fibrosis [53]. *Bifidobacteriaceae* and *Bifidobacterium* are representative probiotics that restore intestinal mucus growth and alleviate intestinal symptoms [54,55]. *Rikenellaceae* and *Alistipes* (butyrate-producing bacteria) exert anti-inflammatory effects by reducing mucosal inflammation [56,57]. *Bacteroidaecea* and *Bacteroides* (propionate-producing bacteria) decrease inflammatory responses by regulating cytokine and lymphocytes [58,59]. Recently, *Faecalibaculum* was identified as a potentially beneficial strain that produces short-chain fatty acids in addition to reducing this colonic inflammation and dysbiosis of the intestinal microflora [60,61]. Our results indicate that the abundance of those bacteria correlates positively with the expression of colonic TJ proteins and negatively with that of colonic pro-inflammatory cytokines and immune response-related genes. Taken together, the data presented herein show that HY7804 improves MASLD symptoms, increases intestinal barrier function, and improves gut microbiota diversity in an MASLD-induced model through the “gut–liver axis”.

## 4. Materials and Methods

### 4.1. Bacterial Culture

*Lactobacillus helveticus* HY7804, isolated from raw fresh milk obtained from domestic animals in the Republic of Korea, was provided by hy Co., Ltd. The *Lactobacillus helveticus* KCTC 3545 type strain was purchased from the Korean Collection for Type cultures (KCTC, Seoul, Republic of Korea), and used as a bacterial control. The two *Lactobacillus helveticus* strains were cultured at 37 °C for 24 h in Man–Rogosa–Sharpe agar (MRS) broth (BD Difco, Sparks, MD, USA) under anaerobic conditions. Fresh cultured strains were freeze-dried and supplied as animal feed.

### 4.2. Animal Experiments

Male C57BL/6 mice (6 weeks old) were obtained from Dooyeol Biotech (Seoul, Republic of Korea) and maintained in cages at 45~65% humidity and a temperature of 21–23 °C under a 12:12 h light/dark cycle. After 7 days of adaptation, the mice were divided randomly to five experimental groups: control (AIN-93G diet, *n* = 7); MASLD-induced diet (MASLD, D09100310, *n* = 7); MASLD diet with pioglitazone (PIO, *n* = 7); MASLD diet with *Lactobacillus helveticus* HY7804 (HY7804, *n* = 7); and MASLD diet with type strain *Lactobacillus helveticus* KCTC 3545 (Type strain, *n* = 7). PIO and probiotics were dissolved in 100 μL of saline and administrated orally at a dose of 10 mg/kg/day [62] and 10^9^ CFU/kg/day [22], respectively, for 7 weeks. The animal study was continued for 7 weeks, during which time food/water intake and body weight were measured weekly. The mice were sacrificed, and blood, liver, epididymal fat, inguinal fat, colon, and cecum tissues were collected. The weight of the liver and fat tissue was measured immediately after extraction. The protocol for the animal experiments protocol was approved by the Institutional Animal Care and Use Committee, hy Co., Ltd., Republic of Korea (IACUC approval number: AEC-2024-0001-Y). In Figure 8 shows the flow chart of the animal tests.

### 4.3. Blood Sample Collection and Biochemical Analysis

The blood samples collected from the abdominal aorta were allowed to stand before serum was obtained by centrifugation for 20 min at 3000× *g* at 4 °C. The serum samples were analyzed to measure AST, ALT, triglyceride, total cholesterol, fasting glucose, LDL-cholesterol, and HDL-cholesterol. Blood biochemical analyses were performed by Dooyeol Biotech (Seoul, Republic of Korea).

### 4.4. H&E Staining and NAS Grading of Liver

The extracted liver tissues were fixed in a 10% (*v*/*v*) formalin solution and embedded in paraffin. The tissue specimens were cut, mounted on slides, and stained with H&E. After H&E staining, the image was scanned by MoticDSAssistant (Motic VM V1 Viewer 2.0). Then, the NAS was assessed by evaluating the levels of hepatic steatosis, lobular inflammation, and hepatocyte ballooning. The NAS are shown in Table 1. Hepatological analysis were performed by Dooyeol Biotech.

### 4.5. RNA Extraction, cDNA Synthesis, and Real-Time PCR Reaction

Total RNA was isolated from the mouse liver and colon tissues using an Easy-spin Total RNA Extraction Kit (iNtRON Biotechnology, Seoul, Republic of Korea). RNA from each sample was reverse transcribed into cDNA at 37 °C for 60 min using the Omniscript RT Kit (Qiagen, Hilden, Germany). The cDNA was amplified using the QuantStudio 6 Flex Real-time PCR System (Applied Biosystems, Waltham, MA, USA) and TaqMan^TM^ Gene Expression Master Mix (Applied Biosystems). The mouse gene-specific probes used for the real-time PCR reactions are shown in Table 2. Gene expression was calculated using the comparative 2^−ΔΔCT^ method. All the data were normalized to that of *Gapdh*.

### 4.6. DNA Extraction and Sample Preparation for Next-Generation Sequencing (NGS)

Total genomic DNA (gDNA) was extracted from the mouse cecum using a DNeasyPowerSoil Kit (Qiagen, Hilden, Germany) and quantified using the Quant-iT PicoGreen^TM^ dsDNA Assay Kit (Invitrogen, Waltham, MA, USA). Library construction was conducted by amplifying the V3-V4 regions of the 16S rRNA genes according to the Illumina 16S Metagenomic Sequencing Library protocols. The first PCR was performed using input DNA, reaction buffer, dNTP mix, each of the universal primers (forward/reverse), and Hercules II fusion DNA polymerase (Agilent Technologies, Santa Clara, CA, USA). The second PCR was conducted using the first PCR product and the Nextera XT Index Kit (Illumina, San Diego, CA, USA). The final PCR amplicons were quantified with KAPA Library Quantification Kits (KAPA Biosystems, Wilmington, MA, USA) and TapeStation D1000 ScreenTape (Agilent Technologies, Waldbronn, Germany). Paired-end sequencing was performed by Macrogen (Seoul, Republic of Korea) using the Illumina MiSeq^TM^ platform (Illumina, San Diego, CA, USA) and the following primers.

V3-Forward: 5′-TCGTCGGCAGCGTCAGATGTGTATAAGAGACAGCCTACGGGNGGCWGCAG-3′,V4-Reverse: 5′-GTCTCGTGGGCTCGGAGATGTGTATAAGAGACAGGACTACHVGGGTATCTAATCC-3′

### 4.7. Amplicon Sequence Variants (ASV) Analysis

Illumina MiSeq raw data for each sample were classified by the Nextera XT index, and paired-end FASTQ files were generated for each sample. The FASTQ files were trimmed to remove the sequencing adaptor sequence and forward/reverse primers from the target gene region using the Curadapt program (v 3.2). Chimera sequences were removed the using consensus method to create ASVs. The DADA2 (v1.18.0) package of the R program (v 4.0.3) was used to correct any errors in the amplicon sequencing process. Sequences with expected errors >2 for paired-end reads were excluded, and data that completed the preprocessing process were subjected to a batch-by-batch error model to remove noise from each sample. After correcting the sequencing errors, the paired-end sequences were assembled into a single sequence, and chimeras were again removed using the DADA2 consensus method to generate ASVs. In addition, for comparative analysis of the microbial communities, the QIIME (v 1.9) program was used to normalize the read count of each sample to the minimum read count among all the samples by applying subsampling. Each ASV sequence was checked against the Reference DB (NCBI 16S Microbial database) through BLAST (v2.9.0) to assign taxonomy information to the subject organism with the highest similarity. If the query coverage of the best hit matching the DB was <85%, or the identity of the matched region was <85%, taxonomy information was not assigned. A bioinformatics analysis was performed using MicrobiomAnalyst (https://www.microbiomeanalyst.ca/, accessed on 2 December 2024) according to the SILVA database. To determine species diversity and evenness of the microbial community within a sample, the Shannon index and Simpson index were calculated to determine α-diversity. β-diversity between samples within the comparison group was calculated based on Bray–Cutis distance, and visualized using PCoA. The correlation between the intestinal microbial community and biomarkers was assessed by calculating Pearson’s coefficient. Correlation heatmaps were visualized using the pheatmap package in the R software (v 4.0.3). All the NGS datasets from this study have been deposited in the NCBI Sequence Read Archive (SRA) under accession code PRJNA1222108.

### 4.8. Statistical Analysis

The animal experiment data are presented as the mean ± standard deviation (SD). One-way ANOVA with Tukey’s post hoc test was used to compare differences between the groups. In vivo test analyses were conducted using the GraphPad Prism 6.0 software (GraphPad Software, San Diego, CA, USA). The ASV analyses were analyzed by a *t*-test/ANOVA using MicrobiomAnalyst. A *p*-value < 0.05 was considered statistically significant.

## 5. Conclusions

The data presented herein suggest that *Lactobacillus helveticus* HY7804 has potential as a functional food supplement that ameliorates MASLD. HY7804 alleviates physiological and histological grades, including the NAS score. Also, HY7804 improves the expression of genes related to hepatic lipogenesis, fatty acid β-oxidation, and apoptosis. The administration of HY7804 restored intestinal barrier function and exerted anti-inflammatory effects by regulating the expression of genes encoding TJ proteins, cytokines, and members of the TLR4/MyD88/NF-κB signaling pathway. Furthermore, HY7804 altered the diversity of the gut microbiota in MASLD-induced mice. Thus, we suggest that HY7804 may be useful as a functional food supplement that ameliorates MASLD symptoms.

## Figures and Tables

**Figure 1 ijms-26-03557-f001:**
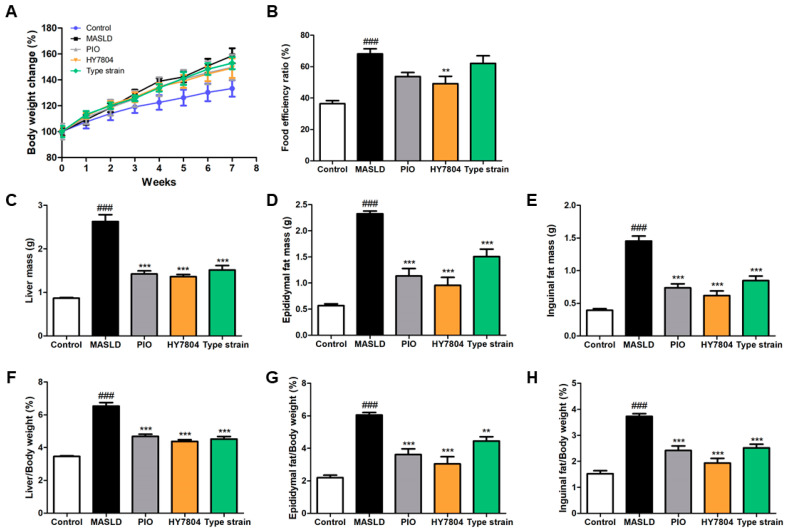
Effects of HY7804 on MASLD symptoms in MASLD-induced mice (*n* = 7 per group): (**A**) changes in body weight, (**B**) FER, (**C**) mass of liver tissues, (**D**) mass of epididymal fat, (**E**) mass of inguinal fat, (**F**) liver/body weight ratio, (**G**) epididymal fat/body weight ratio, and (**H**) inguinal fat/body weight ratio. The data are expressed as the mean ± SD. ^###^
*p* < 0.001 vs. the control group, ** *p* < 0.01, and *** *p* < 0.001 vs. the MASLD group (one-way ANOVA with post hoc analysis). MASLD, mice fed an MASLD-inducing diet; PIO, pioglitazone + MASLD; HY7804, *Lactobacillus helveticus* HY7804 + MASLD; Type strain, *Lactobacillus helveticus* type strain KCTC 3545 + MASLD; FER, food efficiency ratio.

**Figure 2 ijms-26-03557-f002:**
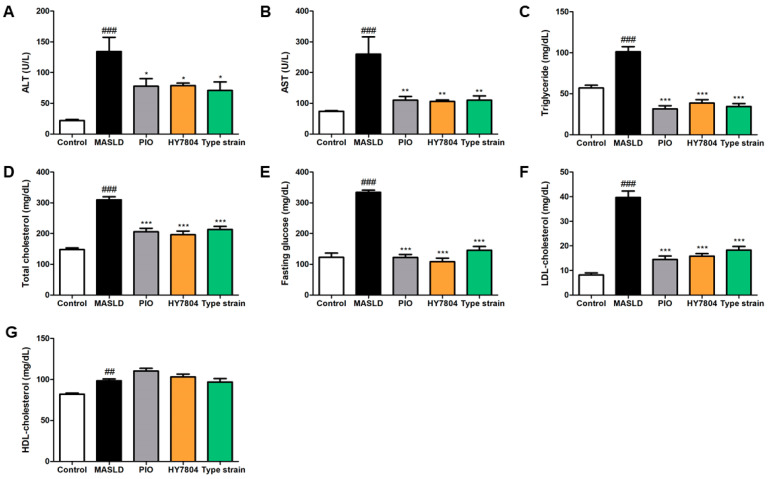
Concentration of biochemical indicators in mouse serum (*n* = 7 per group). (**A**) AST, (**B**) ALT, (**C**) triglycerides, (**D**) total cholesterol, (**E**) fasting glucose, (**F**) LDL-cholesterol, and (**G**) HDL-cholesterol. The data are expressed as the mean ± SD. ^##^
*p* < 0.01 and ^###^
*p* < 0.001 vs. the control group, * *p* <  0.05, ** *p* <  0.01, and *** *p* < 0.001 vs. the MASLD group (one-way ANOVA with post hoc analysis). MASLD, mice fed an MASLD-inducing diet; PIO, pioglitazone + MASLD; HY7804, *Lactobacillus helveticus* HY7804 + MASLD; Type strain, *Lactobacillus helveticus* type strain KCTC 3545 + MASLD; AST, aspartate aminotransferase; ALT, alanine aminotransferase; LDL-cholesterol, low-density lipoprotein cholesterol; HDL-cholesterol, high-density lipoprotein cholesterol.

**Figure 3 ijms-26-03557-f003:**
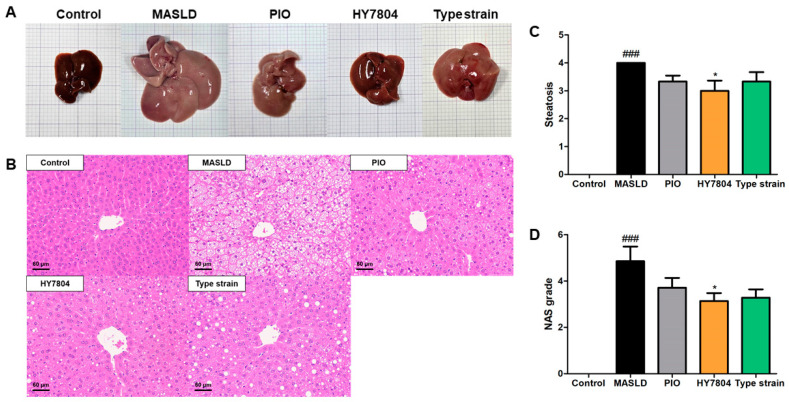
Effects of HY7804 on liver histology (*n* = 7 per group). (**A**) Liver tissue morphology. (**B**) Representative photomicrographs of hepatic histology (magnification ×100). (**C**) Steatosis score. (**D**) NAS. Data are expressed as the mean ± SD. ^###^
*p* < 0.001 vs. the control group, * *p* < 0.05 vs. the MASLD group (one-way ANOVA with post hoc analysis). MASLD, mice fed an MASLD-inducing diet; PIO, pioglitazone + MASLD; HY7804, *Lactobacillus helveticus* HY7804 + MASLD; Type strain, *Lactobacillus helveticus* type strain KCTC 3545 + MASLD.

**Figure 4 ijms-26-03557-f004:**
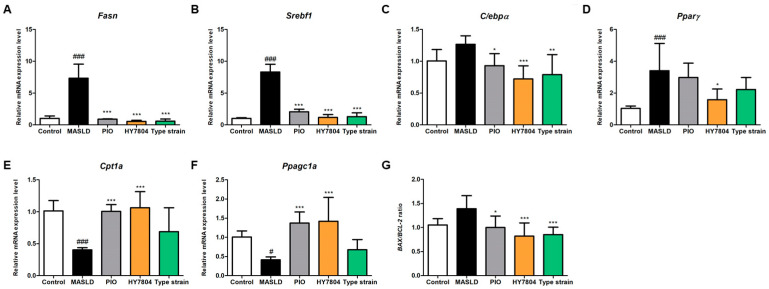
Effects of HY7804 on mRNA expression in the liver of the MASLD-induced mice (*n* = 7 per group). The expression of mRNA encoding genes related to lipogenesis, i.e., (**A**) *Fasn*, (**B**) *Srebf1*, (**C**) *C/ebpα* and (**D**) *Pparγ*. The expression of mRNA encoding genes related to lipid oxidation, i.e., (**E**) *Cpt1a* and (**F**) *Ppargc1a*. (**G**) The expression ratio of *Bax/Bcl-2*. The data are expressed as the mean ± SD. ^#^
*p* < 0.05 and ^###^
*p* < 0.001 vs. the control group; * *p* < 0.05, ** *p* < 0.01, and *** *p* < 0.001 vs. the MASLD group (one-way ANOVA with post hoc analysis). MASLD, mice fed an MASLD-inducing diet; PIO, pioglitazone + MASLD; HY7804, *Lactobacillus helveticus* HY7804 + MASLD; Type strain, *Lactobacillus helveticus* type strain KCTC 3545 + MASLD.

**Figure 5 ijms-26-03557-f005:**
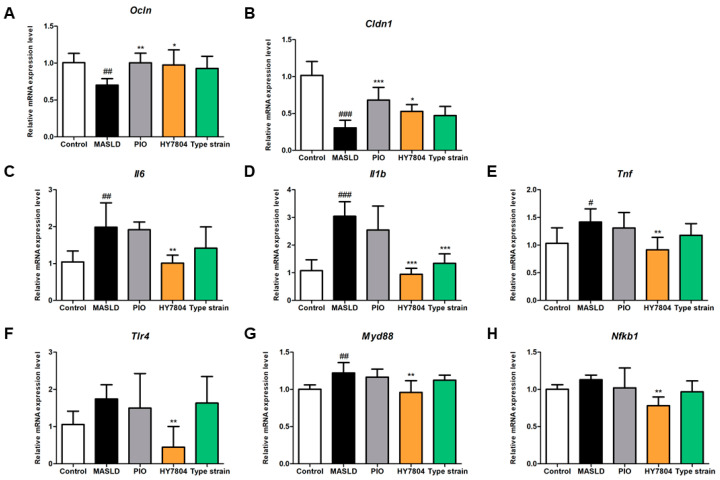
Effects of HY7804 on the expression of mRNAs in the colon of MASLD-induced mice (*n* = 7 per group). The expression of mRNA encoding tight junction proteins (**A**) *Ocln* and (**B**) *Cldn1*. The expression of mRNAs encoding pro-inflammatory cytokines and components of the TLR/MyD88/NF-Κb pathway. (**C**) *Il6*, (**D**) *Il1β*, (**E**) *Tnfα*, (**F**) *Tlr4*, (**G**) *MyD88*, and (**H**) *Nfκb1*. Data are expressed as the mean ± SD. ^#^
*p* < 0.05, ^##^
*p* < 0.01, and ^###^
*p* < 0.001 vs. the control group; * *p* < 0.05, ** *p* < 0.01, and *** *p*  < 0.001 vs. the MASLD group (one-way ANOVA with post hoc analysis). MASLD, mice fed an MASLD-inducing diet; PIO, pioglitazone + MASLD; HY7804, *Lactobacillus helveticus* HY7804 + MASLD; Type strain, *Lactobacillus helveticus* type strain KCTC 3545 + MASLD.

**Figure 6 ijms-26-03557-f006:**
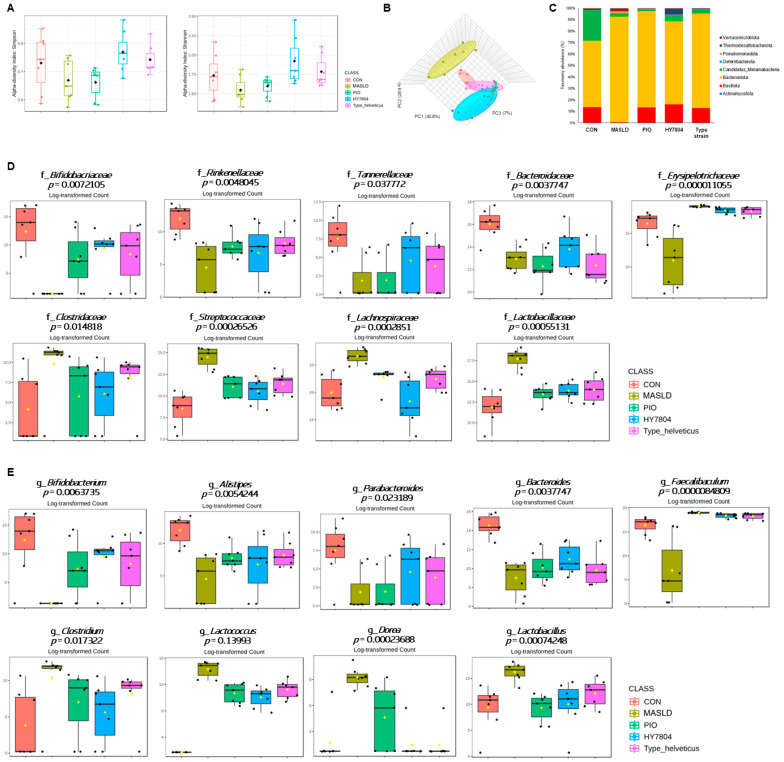
Composition of the gut microbiota in the MASLD-induced mice (*n* = 7 per group). (**A**) α-diversity. (**B**) PCoA of β-diversity using a 3D plot. (**C**) Relative abundance at the phylum level. (**D**) Relative abundance at the family level. (**E**) Relative abundance at genus level. Black dots indicate the values of each samples. Yellow diamonds indicate the mean. Black lines indicate the median. In the graph, red represents CON, yellow represents MASLD, green represents PIO, blue represents HY7804, and purple represents Type_helveticus group. CON, control group mice; MASLD, mice fed an MASLD-inducing diet; PIO, pioglitazone + MASLD; HY7804, *Lactobacillus helveticus* HY7804 + MASLD; Type_helveticus, *Lactobacillus helveticus* type strain KCTC 3545 + MASLD.

**Figure 7 ijms-26-03557-f007:**
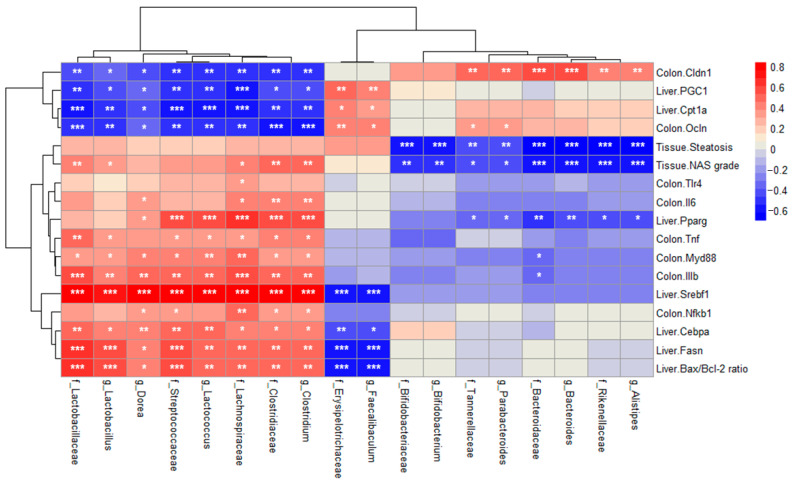
Correlation (Pearson’s correlation coefficient) between the intestinal microbial flora and biochemical indicators. Red indicates a positive Pearson r value and blue indicates a negative Pearson r value. The number indicates the r value. *p*-value < 0.05 was considered statistically significant. (* *p* <  0.05, ** *p* < 0.01, and *** *p* < 0.001).

**Figure 8 ijms-26-03557-f008:**
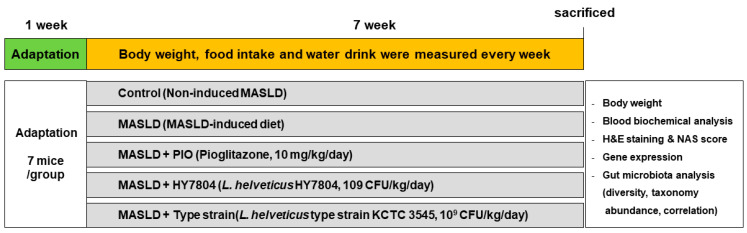
Flow chart of the animal experiments.

**Table 1 ijms-26-03557-t001:** NAS (MASLD Activity Score).

Histological Features	Description	Score
Steatosis	<5%	1
	5–33%	2
	34–65%	3
	>66%	4
Lobular inflammation	No foci	0
	<2 foci /×200 field	1
	2–4 foci/×200 field	2
	>4 foci/×200 field	3
Hepatocyte ballooning	None	0
	Few ballooned cells	1
	Many ballooned cells/prominent ballooning	2

**Table 2 ijms-26-03557-t002:** Gene-specific probes used for real-time PCR reactions.

Gene	Gene Name	Catalog Number
*Gapdh*	Glyceraldehyde-3-phosphate dehydrogenase	Mm99999915_g1
*Fasn*	Fatty acid synthase	Mm00433237_m1
*Pparγ*	Peroxisome proliferator-activated receptor gamma	Mm00440945_m1
*Srebp-1c*	Sterol regulatory element-binding protein 1	Mm00550338_m1
*C/ebpα*	CCAAT/enhancer-binding protein alpha	Mm00514283_m1
*Cpt1a*	Carnitine palmitoyltransferase 1a	Mm01231183_m1
*Ppargc1α*	Peroxisome proliferator-activated receptor gamma coactivator 1-alpha	Mm01208831_m1
*Bax*	BCL2 associated X, apoptosis regulator	Mm00432051_m1
*Bcl-2*	BCL2, apoptosis regulator	Mm00477631_m1
*Il-6*	Interleukin 6	Mm00446190_m1
*Il-1β*	Interleukin 1 beta	Mm00434228_m1
*Tnf* *α*	Tumor necrosis factor	Mm00443258_m1
*Tlr4*	Toll-like receptor 4	Mm00445273_m1
*Nfκb1*	Nuclear factor kappa B subunit1	Mm00476361_m1
*Myd88*	Myeloid differentiation primary response 88	Mm00440338_m1
*Ocln*	Occludin	Mm00500910_m1
*Cldn1*	Claudin-1	Mm01342184_m1

## Data Availability

The data presented in this study are available in the article and Appendix A.

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
