# Peer review of "Lactobacillus helveticus HY7804 Modulates the Gut–Liver Axis to Improve Metabolic Dysfunction-Associated Steatotic Liver Disease in a Mouse Model"

_ijms, 2025, doi:10.3390/ijms26083557_

Round 1

Reviewer 1 Report

Comments and Suggestions for Authors

See attached file for the comments

Comments on the Quality of English Language

It is generally good with some minor revisions (See minor comments)

Author Response

Thank you for your kindly advice which can enhance the quality of our paper.

We believe have resulted in an improved revised manuscript.

We attached response to review file (PDF).

Please confirm our response.

Thank you.

Reviewer 2 Report

Comments and Suggestions for Authors

This manuscript is interesting in studying the effects of Lactobacillus helveticus in the MASLD model in mice. However, I have some suggestions.

1, I noticed that the whole manuscript uses the term NAFLD. However, the term has been changed to MASLD since 2023. Since this will be a publication in 2025, changing the term to MASLD will be more appropriate.

2, Since the author’s focus is on gut-related parameters, using a matched control diet is very important. I noticed the author used AIN-93G as the control diet. Although AIN-93G, a purified diet, is much better than a grain based chow diet, D09100310 was not formulated based on AIN-93G. They used different vitamins and minerals, which might affect your results.

3. On your Figure 1A, the author calculated the body weight change in weight. However, dhe the begining weight is slightly different, calculating the body weight change in percentage can tell us more information. Moreover, Figure 1A and 1B have similar information.

4. For Figure 2E, please indicate the glucose condition. Is it fasting glucose level?

Author Response

(The authors gave the same response as above.)

Round 2

Reviewer 1 Report

Comments and Suggestions for Authors

The comments were addressed well and has improved the paper well.
Just be sure that the comments addressed were reflected on the paper. 

Author Response

We would like to thank you for your comments, which have greatly improved the content of the paper.

We very hope the revised manuscript is accepted for publication in MDPI International Journal of Molecular Sciences.

Thank you.

Yours sincerely,

Joo-Yun Kim, and Hyeonji Kim

Reviewer 2 Report

Comments and Suggestions for Authors

Thanks for revising the manuscript. I have some suggestions for the supplemental file.

  1. In your Figure S1, I would recommend changing the label water drinking/ drink to water intake or water consumption.
  2. for your Figure S2, please add the significance information (p value) in the figure description.

Author Response

1. In your Figure S1, I would recommend changing the label water drinking/ drink to water intake or water consumption.

Answer : Thank you for your kindly recommend. We changed the label of Figure S1-B “water drinking” to “water consumption”. 

2. for your Figure S2, please add the significance information (p value) in the figure description.

 Answer : Thank you for your kindly advice. We added the significance information in the Figure S2.
“Results are presented as the mean ± SD. #p<0.05 vs. Control group, * p < 0.05 vs. MASLD group (one-way ANOVA with post-hoc analysis). MASLD, mice fed an MASLD-inducing diet; PIO, pioglitazone + MASLD; HY7804, Lactobacillus helveticus HY7804 + MASLD; Type strain, Lactobacillus helveticus type strain KCTC 3545 + MASLD.”
